# Gate-reflectometry dispersive readout and coherent control of a spin qubit in silicon

A. Crippa [1], R. Ezzouch [1], A. Aprá [1], A. Amisse[1], R. Laviéville[2], L. Hutin[2], B. Bertrand[2], M. Vinet[2], M. Urdampilleta[3], T. Meunier[3], M. Sanquer[1], X. Jehl[1], R. Maurand[1] & S. De Franceschi [1]

Silicon spin qubits have emerged as a promising path to large-scale quantum processors. In this prospect, the development of scalable qubit readout schemes involving a minimal device overhead is a compelling step. Here we report the implementation of gate-coupled rf reflectometry for the dispersive readout of a fully functional spin qubit device. We use a p-type double-gate transistor made using industry-standard silicon technology. The first gate confines a hole quantum dot encoding the spin qubit, the second one a helper dot enabling readout. The qubit state is measured through the phase response of a lumped-element resonator to spin-selective interdot tunneling. The demonstrated qubit readout scheme requires no coupling to a Fermi reservoir, thereby offering a compact and potentially scalable solution whose operation may be extended above 1 K.

[1] CEA, INAC-PHELIQS, University of Grenoble Alpes, F-38000 Grenoble, France. [2] CEA, LETI, Minatec Campus, F-38000 Grenoble, France. [3] CNRS, Grenoble INP, Institut Néel, University of Grenoble Alpes, F-38000 Grenoble, France. Correspondence and requests for materials should be addressed to A.C. (email: alessandro.crippa@cea.fr) or to R.M. (email: romain.maurand@cea.fr)

The recent years have witnessed remarkable progress in the development of semiconductor spin qubits[1–4] with an increasing focus on silicon-based realizations[5–8]. Access to isotopically enriched $^{28}$Si has enabled the achievement of very long spin coherence times for both nuclear and electron spins[9–11]. In addition, two-qubit gates with increasing high fidelities were demonstrated in electrostatically defined electron double quantum dots[12–14].

While further improvements in single- and two-qubit gates can be expected, growing research efforts are now being directed to the realization of scalable arrays of coupled qubits[15–19]. Leveraging the well-established silicon technology may enable facing the scalability challenge, and initiatives to explore this opportunity are on the way[20]. Simultaneously, suitable qubit device geometries need to be developed. One of the compelling problems is to engineer scalable readout schemes. The present work addresses this important issue.

It has been shown that a microwave excitation applied to a gate electrode drives Rabi oscillations via the electric-dipole spin resonance mechanism[4–6,21–23]. The possibility of using a gate as sensor for qubit readout would allow for a compact device layout, with a clear advantage for scalability. Gate reflectometry probes charge tunneling transitions in a quantum dot system through the dispersive shift of a radiofrequency (rf) resonator connected to a gate electrode[24–27]. Jointly to spin-selective tunneling, e.g. due to Pauli spin blockade in a double quantum dot (DQD), this technique provides a way to measure spin states.

In a similar fashion, the phase shift of a superconducting microwave resonator coupled to the source of an InAs nanowire has enabled spin qubit dispersive readout[22]. In Si, recent gate reflectometry experiments have shown single-shot electron spin detection[28–30].

Here, we combine coherent spin control and gate dispersive readout in a compact qubit device. Two gates tune an isolated hole DQD, and two distinct electric rf tones (one per gate) allow spin manipulation and dispersive readout. Spin initialization and control are performed without involving any charge reservoir; qubit readout relies on the spin-dependent phase response at the DQD charge degeneracy point. We assess hole single spin dynamics and show coherent spin control, validating a protocol for complete qubit characterization exploitable in more complex architectures.

## Results

**Double quantum dot dispersive spectroscopy.** The experiment is carried out on a double-gate, p-type Si transistor fabricated on a silicon-on-insulator 300-mm wafer using an industry-standard fabrication line[6]. The device, nominally identical to the one in Fig. 1c, has two parallel top gates, $G_R$ and $G_C$, wrapping an etched Si nanowire channel. The gates are defined by e-beam lithography and have enlarged overlapping spacers to avoid doping implantation in the channel. The measurement circuit is shown in Fig. 1a. At low temperature (we operate the device at 20 mK using a dilution refrigerator), DC voltages $V_C$ and $V_R$ are applied to these gates to induce two closely spaced hole quantum dots. The control gate $G_C$ delivers also sub-$\mu$s pulses and microwave excitation in the GHz range to manipulate the qubit. The readout gate, $G_R$, is wire-bonded to a 220 nH surface-mount inductor. Along with a parasitic capacitance and the device impedance, the inductor forms a tank circuit resonating at $f_0 = 339$ MHz. Figure 1b shows the phase $\phi$ and attenuation $A$ of the reflected signal as a function of the resonator driving frequency $f_R$. From the slope of the phase trace at $f_0$ we extract a quality factor $Q_{loaded} \simeq 18$. The qubit device acts as a variable impedance load for the resonator, and the resonant

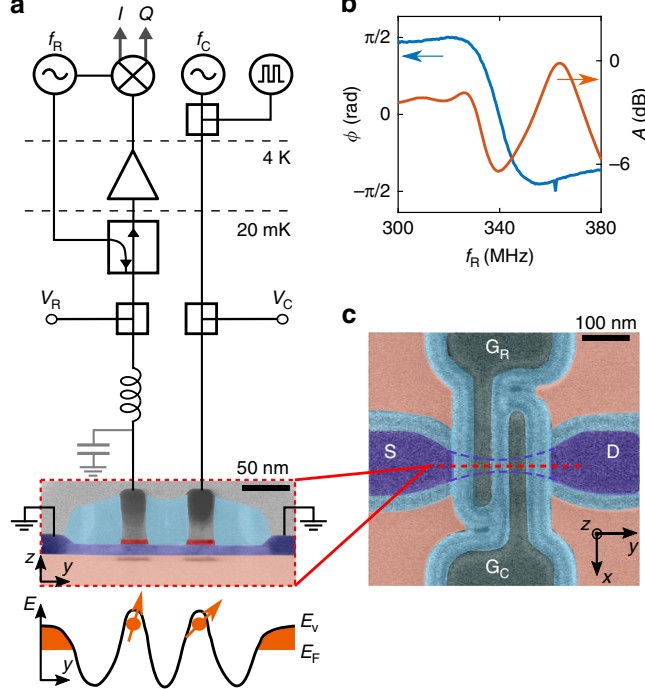

**Fig. 1** Device layout and circuitry for qubit dispersive sensing and manipulation. **a** False-color transmission electron micrograph of a double-gate Si device. The 11-nm-thick, 35-nm-wide Si nanowire (light blue) connects p-type, boron-doped source-drain contacts (dark blue) and lies on a 140-nm-thick SiO$_2$ buffer layer (pink). The two 35-nm-wide gates (grey) are spaced by 35 nm. Si$_3$N$_4$ spacers (cyan) prevent dopant implantation in the Si channel. At 20 mK, proper gate voltages induce the accumulation of two hole quantum dots: one can be used as a spin qubit, the other as a helper dot for qubit readout. One gate is connected to a lumped-element resonator excited at frequency $f_R$ for dispersive readout. A ultra-high frequency digital lock-in demodulates the reflected signal after a directional coupler, separating the incoming and outgoing waves, and a low-noise amplifier at 4 K. The other gate applies square pulses and GHz radiation to drive controlled coherent rotations of the hole spin qubit. At the bottom, DQD energy diagram with $E_v$ as valence band edge and $E_F$ as Fermi energy. **b** Phase response ($\phi$) and attenuation ($A$) of the resonator at base temperature. **c** Scanning electron micrograph of the device

frequency $f_0$ undergoes a dispersive shift according to the state of the qubit.

To determine the charge stability diagram of our DQD, we probe the phase response of the resonator while sweeping the DC gate voltages $V_R$ and $V_C$ (see Supplementary Note 2 and Supplementary Fig. 2).

The diagonal ridge in Fig. 2a denotes the interdot charge transition we shall focus on hereafter. Along this ridge, the electrochemical potentials of the two dots line up enabling the shuttling of a hole charge from one dot to the other. This results in a phase variation $\Delta\phi$ in the reflected signal. Quantitatively, $\Delta\phi$ is proportional to the quantum capacitance associated with the gate voltage dependence of the energy levels involved in the interdot charge transition. Interdot tunnel coupling results in the formation of molecular bonding $(+)$ and anti-bonding $(-)$ states with energy levels $E_+$ and $E_-$, respectively. These states have opposite quantum capacitance since $C_{Q,\pm} = -\alpha^2(\partial^2 E_\pm/\partial\varepsilon^2)$[27]. Here $\varepsilon$ is the gate-voltage detuning along a given line crossing the interdot charge transition boundary, and $\alpha$ is a lever-arm parameter relating $\varepsilon$ to the energy difference between the electrochemical potentials of the two dots (we estimate $\alpha \simeq 0.58$ eV V$^{-1}$ along the detuning line in Fig. 2a). The width of the

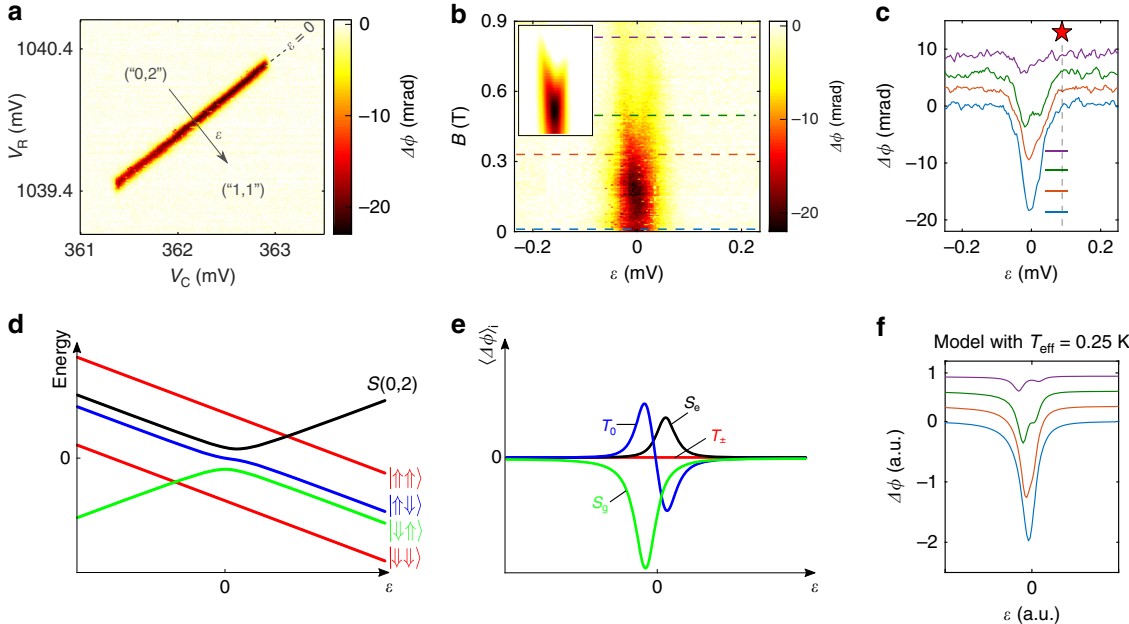

**Fig. 2** Magnetospectroscopy of the double quantum dot. **a** Phase shift of the reflected signal as a function of $V_C$ and $V_R$ near the interdot transition line under study. The arrow indicates $\varepsilon$ detuning axis. **b** Interdot dispersive signal as a function of a magnetic field $B$ oriented along the nanowire axis. The phase response diminishes with $B$, denoting a interdot charge transition of $(0, 2) \leftrightarrow (1, 1)$ type. Inset: theoretical prediction of the dispersive response for a DQD model taking into account thermal spin populations, see Supplementary Note 3. **c** Line cuts of the plot in panel b) at the position of the dashed lines. Data are offset for clarity. **d** Schematic of the DQD energy levels close to a $(0, 2) \leftrightarrow (1, 1)$ transition at finite $B$ and for $|g_L^* - g_R^*| = 0.5$. **e** Thermally-averaged phase response $\langle \Delta\phi \rangle_i$ with $T_{\text{eff}} = 0.25$ K. $\langle \Delta\phi \rangle_i$ is second derivative of the energy-level dispersion of state $i$ in d), weighted by the occupation probability. Here $i$ labels the different DQD states, i.e. the singlets $S_g$ (green) and $S_e$ (black), and the triplets $T_0$ (blue), $T_-$ (red), and $T_+$ (red). **f** Qualitative phase shift resulting from the sum of all $\langle \Delta\phi \rangle_i$ from **e**. A double-peak structure emerges at sufficiently high $B$ in qualitative agreement with the experimental data in **c**)

$\Delta\phi$ ridge, once translated into energy, gives the interdot tunnel coupling, $t$. We estimate $t$ between 6.4 and 8.5 μeV, depending on whether thermal fluctuations contribute or not to the dispersive response (see Supplementary Note 3).

The total charge parity and the spin character of the DQD states can be determined from the evolution of the interdot ridge in an applied magnetic field, $B$[31]. Figure 2b shows the $B$-dependence of the phase signal at the detuning line indicated in Fig. 2a. Four representative traces taken from this plot are shown in Fig. 2c. The interdot phase signal progressively drops with $B$. At $B = 0.355$ T the line profile is slightly asymmetric, while a double-peak structure emerges at $B = 0.46$ T. The two peaks move apart and weaken by further increasing $B$, as revealed by the trace at $B = 0.85$ T.

The observed behavior can be understood in terms of an interdot charge transition with an even number of holes in the DQD, in a scenario equivalent to a $(0, 2) \leftrightarrow (1, 1)$ transition. We shall then refer to a "$(0, 2)$" and a "$(1, 1)$" state, even if the actual number of confined holes is larger (we estimate around ten, see Supplementary Note 2). The $\varepsilon$ dependence of the DQD states at finite $B$ is presented in Fig. 2d. Deeply in the positive detuning regime, different $g$-factors for the left ($g_L^*$) and the right dot ($g_R^*$) result in four non-degenerate $(1, 1)$ levels corresponding to the following spin states: $|\Downarrow\Downarrow\rangle$, $|\Uparrow\Downarrow\rangle$, $|\Downarrow\Uparrow\rangle$, $|\Uparrow\Uparrow\rangle$[22,32,33]. At large negative detuning, the ground state is a spin-singlet state $S(0,2)$ and the triplet states $T(0,2)$ lie high up in energy. Around zero detuning, the $|\Uparrow\Downarrow\rangle$, $|\Downarrow\Uparrow\rangle$ states hybridize with the $S(0, 2)$ state forming an unpolarized triplet $T_0(1, 1)$ and two molecular singlets, $S_g$ and $S_e$, with bonding and anti-bonding character, respectively (Supplementary Note 3).

We use the spectrum of Fig. 2d to model the evolution of the interdot phase signal in Fig. 2b, c. Importantly, we make the assumption that the average occupation probability of the available excited states are populated according to a Boltzmann distribution with an effective temperature $T_{\text{eff}}$, which is used as a free parameter. Because the reflectometry signal is averaged over many resonator cycles, $\Delta\phi = \sum_i \langle \Delta\phi \rangle_i$, where $\langle \Delta\phi \rangle_i$ is the phase response associated to state $i$ weighted by the respective occupation probability[31] (here $i$ labels the DQD levels in Fig. 2d). Figure 2e shows $\langle \Delta\phi \rangle_i$ as a function of $\varepsilon$ for $T_{\text{eff}} = 250$ mK. The spin polarized triplet states $T_-$ and $T_+$ (i.e. $|\Downarrow\Downarrow\rangle$ and $|\Uparrow\Uparrow\rangle$, respectively) are linear in $\varepsilon$ and, therefore, they do not cause any finite phase shift; $S_g$, $S_e$, and $T_0(1, 1)$, on the other hand, possess a curvature and are sensed by the reflectometry apparatus (Supplementary Note 3). We note that the phase signal for $T_0(1, 1)$ has a peak-dip line shape whose minimum lies at positive $\varepsilon$ (blue trace), partly counterbalanced by the positive phase signal due to $S_e$. The $S_g$ state causes a pronounced dip at negative $\varepsilon$ (green trace), dominating over the peak component of $T_0$. The overall net result is a phase signal with an asymmetric double-dip structure consistent with our experimental observation.

This simple model, with the chosen $T_{\text{eff}} = 250$ mK, qualitatively reproduces the emergence of the double-dip structure at $B \sim 0.4$ T, as well as its gradual suppression at higher $B$, as shown in the inset to Fig. 2b and in Fig. 2f (increasing the Zeeman energy results in the depopulation of the $S_g$ and $T_0$ excited states in favor of the $T_-(1,1)$ ground state, for which $\Delta\phi = 0$).

**Dispersive detection of electric-dipole spin resonance**. Now that we have elucidated the energy-level structure of the DQD, we can discuss the operation of the device as a single-hole spin qubit with electrical control and dispersive readout. Electric-dipole spin resonance (EDSR)[6,23,34] is induced by a microwave voltage modulation applied to gate $G_C$. To detect EDSR dispersively, the resonating states must have different quantum capacitances.

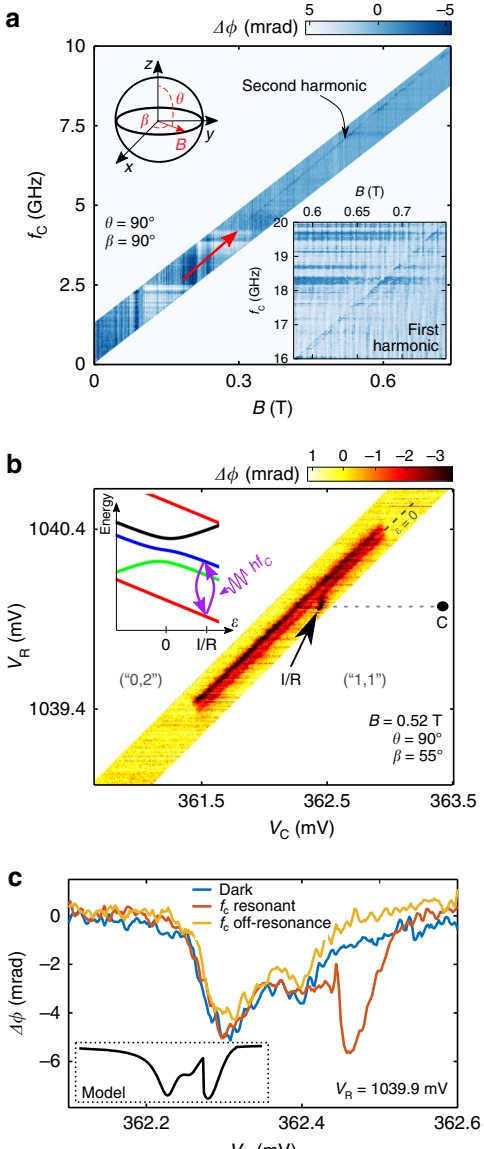

**Fig. 3** Experimental detection of electric-dipole spin resonance (EDSR). **a** Phase response as a function of $B$ and microwave frequency $f_C$. $B$ is oriented along the $y$ direction with respect to the frame of Fig. 1a. The linear phase ridge denoted by a red arrow is a characteristic signature of EDSR. It corresponds to a second-harmonic signal, while the much weaker first harmonic is shown in the lower inset. **b** Stability diagram at $B = 0.52$ T (orientation $\beta = 55°$ and $\theta = 90°$ according to the diagram in upper inset of a) with $f_C = 7.42$ GHz and microwave power $P_C \approx -80$ dBm. EDSR between $T_-(1, 1)$ and $T_0(1, 1)$ (purple arrows in inset) is driven at point $I$. In the stability diagram, the change of population induced by EDSR is visible as a localized phase signal at point I/R. **c** Phase shift at $V_R = 1039.9$ mV as a function of $V_C$ without microwave irradiation (dark), and under on-resonance and off-resonance excitation at $f_C = 7.42$ and 7.60 GHz, respectively. EDSR-stimulated transitions appear as a pronounced peak whose position and line shape are compatible with our model (inset)

The DQD is initially tuned to the position of the red star in Fig. 2c, where the DQD is in a "shallow" (1,1) configuration, i.e. close to the boundary with the (0,2) charge state (more details in Supplementary Note 4 and Supplementary Fig. 4).

Figure 3a shows the dispersive measurement of an EDSR line. The microwave gate modulation of frequency $f_C$ is applied continuously and $B$ is oriented along the nanowire axis.

We ascribe the resonance line to a second-harmonic driving process where $2hf_C = g\mu_B B$ ($h$ the Planck's constant, $\mu_B$ the Bohr magneton and $g$ the effective hole $g$-factor). From this resonance condition we extract $g = 1.735 \pm 0.002$, in agreement with previous works[6,23]. The first harmonic signal, shown in the inset to Fig. 3a, is unexpectedly weaker. Though both first and second harmonic excitations can be expected[35], the first harmonic EDSR line (inset to Fig. 3a) is unexpectedly weaker. A comparison of the two signal intensities requires the knowledge of many parameters (relaxation rate, microwave power, field amplitude) and calls for deeper investigations.

The visibility of the EDSR signal can be optimized by a fine tuning of the gate voltages. Figure 3b shows a high-resolution measurement over a narrow region of the stability diagram around the interdot charge transition boundary at $B = 0.52$ T; the interdot line has a double-peak structure, consistently with the data in Fig. 2b, c. The measurement is performed while applying a continuous microwave tone $f_C = 7.42$ GHz. EDSR appears as a distinct phase signal around $V_C \simeq 362.5$ mV and $V_R \simeq 1040$ mV, i.e. slightly inside the (1,1) charge region, pinpointed by the black arrow as I/R. Such EDSR feature is extremely localized in the stability diagram reflecting the gate-voltage dependence of the hole $g$-factor[23].

Figure 3c displays line cuts across the interdot transition line at fixed $V_R$ and different microwave excitation conditions. With no microwaves excitation, we observe the double-peak line shape discussed above. With a microwave gate modulation at $f_C = 7.42$ GHz, the spin resonance condition is met at $V_C \simeq 362.45$ mV, which results in a pronounced EDSR peak, the same observed at point I/R in Fig. 3b (see also Supplementary Fig. 4). The peak vanishes when $f_C$ is detuned by 20 MHz (cyan trace).

At point I/R, resonant microwave excitation enables the spectroscopy of the $T_0(1, 1)$ state. The inset to Fig. 3c shows the signal we expect from our model (Supplementary Note 4). In a small detuning window, the populations of $T_-(1, 1)$ and $T_0(1, 1)$ are assumed to be balanced by EDSR (see the energy levels in the inset to Fig. 3b); this results in a phase signal dramatically enhanced resembling the feature centered at I/R in the main panel. A further confirmation that the spin transitions are driven between $T_-(1, 1)$ and $T_0(1, 1)$ is given by the extrapolated intercept at 0 T of the EDSR transition line in Fig. 3a, found much smaller (<100 MHz) than $t$. In the following, we shall use point I/R to perform qubit initialization and readout.

**Qubit control and readout.** The device is operated as a spin qubit implementing the protocol outlined in Fig. 4a. The voltage sequence in the upper part of Fig. 4a tunes the DQD at the control point C ($\simeq 1$ mV deep in the (1, 1) region) where holes are strongly localized in either one or the other dot with negligible tunnel coupling. A microwave burst of duration $\tau_{burst}$ and frequency $f_C$ drives single spin rotations between $|\Downarrow\Downarrow\rangle$ and $|\Uparrow\Downarrow\rangle$; the system is then brought back to I/R in the "shallow" (1, 1) regime for a time $t_{wait}$ for readout and initialization. The dispersive readout eventually relies on the spin-resolved phase shift at I/R, though the reflectometry tone $f_R$ is applied during the whole sequence period $T_M$ and the reflected signal is streamed constantly to the acquisition module.

First, we determine the lifetime $T_1$ of the excited spin state at the readout point I/R by sweeping $t_{wait}$ after a $\pi$-burst at point C. The results are shown in Fig. 4b. The phase signal rapidly diminishes with increasing $t_{wait}$ because spin relaxation depopulates the excited spin state in favor of the non-dispersive $T_-(1, 1)$ ground state. The estimated spin lifetime at the readout position is $T_1 = 2.7 \pm 0.7$ μs (see Supplementary Note 5). By shifting the position of a 100 ns microwave burst within a 12 μs pulse, no

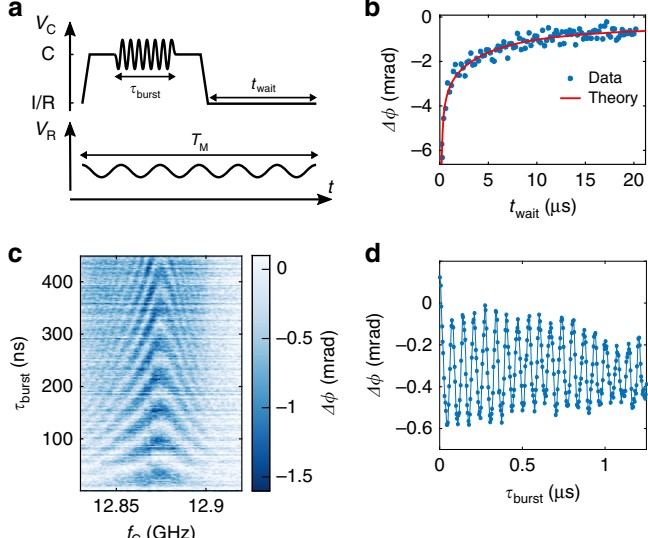

**Fig. 4** Single spin control and dispersive sensing. **a** The pulse sequence alternating between "deep" (1, 1) regime (C) for spin manipulation and "shallow" (1, 1) regime (I/R), close to the (0, 2) ↔ (1, 1) transition, for the readout and resetting of the spin system. A microwave burst rotates the hole spin during the manipulation stage. The readout tone is continuously applied throughout the sequence period $T_M$. **b** Phase shift as a function of $t_{wait}$ for a ≃1 mV pulse on $V_C$ with $τ_{burst} = 100$ ns and $f_C = 12.865$ GHz, with $B = 0.512$ T along $β = 0°$ and $θ = 60°$. The phase signal approaches 0 when $t_{wait} \gg T_1$. A simple model yields $T_1 = 2.7 ± 0.7$ μs. **c** Dispersive signal $Δϕ$ ($f_C$, $τ_{burst}$), measured with the detuning pulses of **a**) with $t_{wait} = 1$ μs. Four maps have been averaged. **d** Phase response as a function of EDSR burst time at $f_C = 12.865$ GHz. The plot shows Rabi oscillations with 15 MHz frequency due to coherent spin rotations. Each data point is integrated for 100 ms and then averaged over 30 traces

clear decay of the dispersive signal is observed, which suggests a spin lifetime at manipulation point longer than 10 μs.

We demonstrate coherent single spin control in the chevron plot of Fig. 4c. The phase signal is collected as a function of microwave burst time $τ_{burst}$ and driving frequency $f_C$. The spin state is initialized at point I/R ($t_{wait} \sim T_1$). In Fig. 4d the phase signal is plotted as a function of $τ_{burst}$ with $f_C$ set at the Larmor frequency $f_{Larmor}$. The Rabi oscillations have 15 MHz frequency, consistent with Refs. roro and gtensor. The non-monotonous envelope is attributed to random phase accumulation in the qubit state by off-resonant driving at $f_{Larmor} ± f_R$ due to up-conversion of microwave and reflectometry tones during the manipulation time. Data in Fig. 4d have been averaged over 30 measurements though the oscillations are easily distinguishable from single scans where each point is integrated over 100 ms. Figure 4 witnesses the success of using electrical rf signals both for coherent manipulation by EDSR and for qubit-state readout by means of gate reflectometry.

## Discussion

The measured $T_1$ is compatible with the relaxation times obtained for hole singlet-triplet states in acceptor pairs in Si[36] and in Ge/Si nanowire double quantum dots[37]; in both cases $T_1$ has been measured at the charge degeneracy point with reflectometry set-ups similar to ours. Nonetheless, charge detector measurements have shown $T_1$ approaching 100 μs for single hole spins in Ge hut wire quantum dots[38] and $\lesssim 1$ ms for Ge/Si singlet-triplet systems[39]. This suggests that despite the intrinsic spin-orbit coupling single spin lifetimes in the ms range might be achievable in Si too.

Strategies to boost $T_1$ at the readout point may consist of inserting rf isolators between the coupler and the amplifier to reduce the backaction on the qubit and avoiding high-κ dielectric in the gate stack to limit charge noise.

We note that $T_1$ could depend on the orientation of the magnetic field as well[40]. Future studies on magnetic field anisotropy will clarify whether $T_1$, along with the effective g-factors (and hence the dispersive shift for readout) and Rabi frequency, can be maximized at once along a specific direction. Technical improvements intended to enhance the phase sensitivity, like resonators with higher Q-factor and parametric amplification, could push the implemented readout protocol to distinguish spin states with a micro-second integration time, enabling single shot measurement as reported in a recent experiment with a gate-connected superconducting resonant circuit[41]. Lastly, the resonator integration in the back-end of the industrial chip could offer the possibility to engineer the resonant network at a wafer scale, guaranteeing controlled and reproducible qubit-resonator coupling.

The gate-based dispersive sensing demonstrated here does not involve local reservoirs of charges or embedded charge detectors. This meets the requirements of forefront qubit architectures (e.g. Ref. [16]), where the spin readout would be performed at will by any gate of the 2D quantum dot array by frequency multiplexing.

Dispersive spin detection by Pauli blockade has a fidelity not constrained by the temperature of the leads. As recently shown[42], isolated DQDs can serve as spin qubits even if placed at environmental temperatures exceeding the spin splitting, like 1 K or more. This should relax many cryogenic constraints and support the co-integration with classical electronics, as required by a scale-up perspective[19].

## Methods

**Device fabrication**. The fabrication process of the device was carried out in a 300 mm CMOS platform and is described in Ref. [6].

**Experimental set-up**. The experiment is performed by exciting the resonator input at $f_R = f_0 = 339$ MHz and power $P_R \approx -110$ dBm. We measure the phase variation $Δϕ$ of the reflected signal isolated from the incoming wave by a directional coupler, amplified by 35 dB at 4 K and demodulated to baseband using homodyne detection. The complete circuit diagram of the experimental setup for qubit manipulation and dispersive readout is provided in Supplementary Note 1 and Supplementary Fig. 1.

## Data availability

The data that support the findings of this study are available from the corresponding author upon reasonable request.

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

## Acknowledgements

We thank K.D. Petersson, M.L.V. Tagliaferri, M.F. Gonzalez-Zalba and Y.-M. Niquet for fruitful discussions. The work was supported by the European Union's Horizon 2020 research and innovation program under Grant Agreement No. 688539 MOS-QUITO (http://mos-quito.eu) and by the ERC Project No. 759388 LONGSPIN.

## Author contributions

A.C. and R.E. performed the experiments with help from R.M., A.C., R.M., and S.D.F. designed the experiment. R.L., L.H., B.B., M.V. fabricated the sample. A.C. analyzed the results with inputs from R.E., A. Aprá, A. Amisse, M.U., T.M., M.S., X.J., R.M., and S.D.F., A.C., R.M., and S.D.F. wrote the paper. M.S., X.J., M.V., and S.D.F. initiated the project.

## Additional information

**Competing interests:** The authors declare no competing interests.

