## [Peer Review File · Nature Communications]

Reviewers' comments:

Reviewer #1 (Remarks to the Author):

This paper by Crippa et al. reports the first experimental demonstration (to my knowledge) in a semiconductor device of the coherent operation of a spin qubit combined with measurement using a gate-based dispersive readout technique (GDRO). GDRO is considered by many in the solid-state qubit community to be an important (and maybe essential) resource for future large-scale QC implementations, because it requires only a single wire/electrode to measure a qubit state, in contrast with more complex multi-terminal devices (such as SETs), thus reducing the wiring complexity of large 2D qubit arrays.

There has been considerable work from a number of groups worldwide devoted to demonstrations of GDRO in recent years, and very recently there have been the first demonstrations of single-shot spin readout using GDRO. However, none of these demonstrations, to my knowledge, have combined GDRO with coherent qubit control. [If I have missed this, then I would encourage the authors to point it out in their manuscript, and reference accordingly.] The combination of GDRO with coherent control makes the current work an important experimental demonstration, which will be of considerable interest to the spin qubit community.

The qubit system studied is based on a CMOS-compatible silicon nanowire double (hole) quantum dot device, where the hole quantum dots are formed using single wrap-around gates, and readout is achieved using the dispersive reflected rf signal from the gate defining one of the dots. The other gate electrode is used to apply a different rf tone, which enables electrically-driven spin resonance (EDSR) of the (many) hole spin qubit confined under that gate.

The experimental data in the paper is convincing, and the authors provide a convincing theoretical model (taking into account thermal spin populations), which shows excellent consistency with the form of the dispersive readout signal that is obtained. The paper is also clearly written and well organised. Given the quality of the work, and the significance of these novel experimental results (noted above), I believe that the paper is certainly suitable for publication in Nature Communications, with some small modifications, as noted below.

Here are my comments/questions:

1. The current title reads:

"Gate-reflectometry dispersive readout of a spin qubit in silicon".

Given that GDRO of a spin qubit has been achieved by a number of groups in the past, I would suggest that the title should better reflect the novelty of this result, which is to combine GDRO with coherent spin control for the first time. Therefore, perhaps a better title might be:

"Gate-reflectometry dispersive readout and control of a spin qubit in silicon". ??

2. The double dot system is operated between "(0,2)" to "(1,1)" states, with the actual number of holes not known accurately. The authors are up-front about this, and I do not see this as a problem, since it doesn't detract from the significance of the result. However, the authors should include a charge stability map (extending that in Fig. 2a) showing the nearby charge states, so that the reader can assess for themselves the level of disorder in the system. Such a plot could be presented in the Supplementary Material.

3. I could not find any mention (in either the caption or text) of what the data in the inset to Figure 2b was meant to be. Did I miss this? If so, could the authors point out where they describe it? If not, they should add this explanation to the revised manuscript.

4. On Page 3, the authors write: "From this resonance condition we extract $g = 1.735 \pm 0.002$." Could the authors comment on whether this value is expected to a hole QD in such a system,

maybe by comparing with previous work?

5. Immediately afterwards, they state: "The first harmonic signal, shown in the inset to Fig. 3a, is unexpectedly weaker."

Do the authors have any explanation/speculation as to why? It would be good to comment on this in the revised manuscript.

6. In the discussion of the coherent spin control protocol on Page 4 and in Figures 3 & 4, I found it strange/confusing that they labelled the coherent control point as "M", which to me indicates "Measure" point. It would be much clearer if they relabelled "M" as "C", for "Control". The point "I" can stay as "I", or could be "I/M", for "Initialize/Measure".

7. Could the authors please comment on why the Rabi oscillations in Figure 4d have such a strange looking envelope? Why do they not appear to decay in a more typical manner?

8. On a related note, in Figure 4d, why did they stop taking data at 1.2 us burst length? What does the data look like up to 2 microseconds? This would give a clearer idea of the decoherence mechanism.

9. Why did they not do a Ramsey sequence to measure T_2^* ? This seems an obvious experiment. Can they comment on this?

10. Do the authors believe that T_1 processes ultimately limit coherence in this experiment? If so, they should also state this.

Finally, I list here a few minor suggestions, typos or grammatical errors that I picked up:

a. Page 1, Column 2, line 2:

"combined to .." should read "combined with .."

b. In Figure 1, the energy diagram should include labels for E_v and E_F .

c. Page 3, Column 1, Line 2:

"deep" should read "dip"

d. Page 3, Column 2, 3rd-last Line:

"consistently" should read "consistent"

e. Page 4, Column 2, 3rd Paragraph:

"supposed balanced" would better read as "assumed to be balanced"

Reviewer #2 (Remarks to the Author):

This work advances quantum CMOS by implementing dispersive spin readout. Incorporating both scalable control and readout into a functioning qubit built from a semiconductor manufacturing platform, which is also scalable, is an important technological advance in quantum technology, an advance that should be of interest to the wider field.

Conceptually, the followed approach is very similar to the work of Reference 22 where a coplanar waveguide resonator was used instead of the lump-element resonator to achieve essentially the same, i.e. dispersive spin readout by spin blockade in a double quantum dot. This conceptual link with the state-of-the-art is not well explained, giving a somewhat false impression about the novelty of the work. One novelty resides in the deeper understanding of the physics at play the

work provides, e.g. the interplay between the possible spin states on the overall quantum capacitance. This is a very convincing aspect of the work, yet the authors are not transparent about what their contribution to the dispersive spin readout really is.

Emphasizing on this new understanding, could the authors offer a discussion on the prospect of single-shot readout for their unique spin qubit platform, i.e. what is missing to get there in terms of quality factor, monolithic integration of the lumped-element resonator, noise level, etc.?

The authors may consider the following other questions in improving further the manuscript

1. Would spin-flip tunneling terms in the model Hamiltonian change the overall interpretation? If yes, why can these processes be neglected given that they are another possible effect of the spin-orbit coupling?
2. The electronic temperature is stated to be 250 mK, corresponding to an energy of 22 micro electron-volt. How was the smaller tunneling coupling energy deconvoluted from the thermal broadening when using the data of Figure 2a? In usual charge sensing, the inter-dot charge transition always has a temperature broadening associated to it.
3. I can appreciate the hard work involved but why not showing the EDSR resonance of the other dot?
4. In what conditions removing the reservoirs would enable higher operating temperature? This statement is repeated twice but not explained nor referenced.

Reviewer #3 (Remarks to the Author):

Silicon spin qubits are now one of the leading quantum computer technologies, but share with other technologies the challenge of scaling to large circuits. The authors' CMOS-based design is promising in this respect. In Ref. [6], they showed nicely that a CMOS double dot works as a hole spin qubit, measured using the electrical current through the device. As is well known, this readout method does not allow scaling up.

In this new paper, the authors apply fairly established gate-reflectometry techniques to the same kind of device. This is an important step forward, because (i) it means an array of qubits can be operated without coupling to reservoirs, (ii) it allows for single-shot readout and therefore computational error correction. The paper shows how the different spin states contribute to the readout signal, which is also applicable to other double-dot devices. It also combines reflectometry with spin resonance to coherently manipulate spin states (i.e. to realise a one-qubit gate), and to measure the spin lifetime and coherence time.

The experiment is carefully done and well explained, including the theory part in figure 2. In deciding whether it deserves publication in Nature Comms, the question is whether it goes far enough beyond Ref 6 and Refs 28-29. Ref 6 already contains the device technology and the qubit control, and Refs 28-29 use essentially the same readout scheme in other silicon qubit realizations – in fact, those references go further by actually achieving single-shot sensitivity.

It is a fine judgement, but on balance my opinion is that the work still merits high-impact publication. The reason is that it combines, in principle, a scalable device fabrication with a scalable readout technology, and shows that they work together. It is therefore likely to be of broad interest to researchers in the entire field of spin quantum computing.

I have some detailed technical comments:

- Both the abstract and the conclusion claim that operation "may be extended well above 1 Kelvin". This claim should be justified or removed, especially since the authors apparently decided not to test it directly by warming their cryostat. Anyone copying this technique would no longer need to worry about the reservoir temperature, but the readout might still be affected by excited charge states, phonons, other spin relaxation mechanisms, etc.
- The magnetic field direction in Fig. 2 is not clear.
- The second harmonic driving process in Fig. 3 is indeed surprising. Is it possible that the strong

line is in fact the fundamental? I do not know whether such a large g-factor is plausible in this material. There is a paper (PRL 115, 106802) that shows how to tell by measuring the phase dependence in a Ramsey experiment.

-Some small errors:

-- Figure 2 caption, last line: an "at" seems missing.

--Something is hiding behind the colour scale in Fig. 3b.

--Page 3: "deep"->"dip".

Reply to Referee Report of the manuscript

Gate-reflectometry dispersive readout and coherent control of a spin qubit in silicon

by A. Crippa et al.

We thank Editor and Reviewers for consideration and careful analysis of our manuscript.

In the following we reply to the questions and criticisms arisen by the three Reviewers.

Revised and new sentences are referenced in the following document with line numbers of the manuscript and are highlighted in red in the manuscript text.

Reviewer #1

We reply to her/his remarks as follows.

At first, we would like to state that she/he is right in the preface: no combination of gate-based dispersive readout (GDRO) and coherent qubit operations has been showed jointly so far.

1. The current title reads:

“Gate-reflectometry dispersive readout of a spin qubit in silicon”.

Given that GDRO of a spin qubit has been achieved by a number of groups in the past, I would suggest that the title should better reflect the novelty of this result, which is to combine GDRO with coherent spin control for the first time. Therefore, perhaps a better title might be: “Gate-reflectometry dispersive readout and control of a spin qubit in silicon”. ??

We see the point arisen by the Reviewer. The title of the paper has been modified accordingly, and now reads:

“Gate-reflectometry dispersive readout and coherent control of a spin qubit in silicon”.

2. The double dot system is operated between “(0,2)” to “(1,1)” states, with the actual number of holes not known accurately. The authors are up-front about this, and I do not see this as a problem, since it doesn’t detract from the significance of the result. However, the authors should include a charge stability map (extending that in Fig. 2a) showing the nearby charge states, so that the reader can assess for themselves the level of disorder in the system. Such a plot could be presented in the Supplementary Material.

We accede to the Reviewer’s request. The new Figure S2 in the Supplementary Material shows a larger portion of the gate voltage space spanned by V_C and V_R . Comments to this figure are reported in Supplementary Note 2 of Supplementary Material and here below:

Supplementary Figure 2. Dispersively detected charge stability diagram of the device as a function of the two top gate voltages, V_C and V_R . In the bottom panel, both gates are tuned in the many hole regime, characterized by a relatively regular arrangement of the interdot transition lines. In the upper panel, V_R approaches the gate voltage room temperature threshold. In this regime the stability diagram is more disordered. The blue square denotes the area zoomed-in in Fig. 2a of the main text.

“Supplementary Figure S2 shows two stability diagrams of the device under investigation. Both plots share the same V_C voltage range. The other gate tunes the electrostatics of the channel from the many hole regime (bottom panel) to the voltage region we use to implement our qubit (top panel). In particular, the blue square highlights the area zoomed in the stability map of Fig. 2a in the main text.

Considering the diagram as a whole, two sets of features are present. First, a series of nearly horizontal parallel lines are visible. These lines repeat quite regularly from metallic DQDs to depletion, even the silicon channel is completely closed (data not shown). Consequently, we speculate that these features are related to the charging of objects extrinsic to the channel.

On top of this background, most of the short diagonal cuts on the yellow background are interdot transition lines.

The bottom part of Fig. S2 reports the many hole regime where the voltage spacing between DQDs is approximately constant. The typical gate voltage between two charge states is about 25 mV. This value is consistent with other experiments on similar samples [1, 2].

Out of the many hole regime, the interdot lines are unevenly spaced, as displayed in the top panel. Importantly, for interdot tunnel couplings of few GHz (like the one studied in the main text), the interdot transition lines are quite thin in gate voltage, and are very likely not resolved in large maps obtained with large voltage steps.

We use the threshold voltages at room temperature of the two gates and the addition voltage of the many hole regime for a rough estimation of the absolute filling of the dots. We obtain an order of magnitude of 5 holes and 10-20 holes in the left (i.e. mainly controlled by V_R) and right dot (mainly controlled by V_C), respectively.”

3. I could not find any mention (in either the caption or text) of what the data in the inset to Figure 2b was meant to be. Did I miss this? If so, could the authors point out where they describe it? If not, they should add this explanation to the revised manuscript.

We apologize for the oversight. We have added a reference to the inset of Fig. 2b in the text (line 190) and we have inserted the following caption in the figure:

“Inset: theoretical prediction by a DQD model taking into account thermal spin populations, see Supplementary Note 3.”

4. On Page 3, the authors write: “From this resonance condition we extract $g = 1.735 \pm 0.002$.”

Could the authors comment on whether this value is expected to a hole QD in such a system, maybe by comparing with previous work?

In previous papers (Phys. Rev. Lett. **120**, 137702 (2018) and Nano Lett. 16, 88–92 (2016)) we measured the g factor anisotropy in similar devices. The g-factor 1.735 is a value in the typical range we are used to measure. We have added a remark on that point in the revised manuscript at line 216.

5. Immediately afterwards, they state: “The first harmonic signal, shown in the inset to Fig. 3a, is unexpectedly weaker.”

Do the authors have any explanation/speculation as to why? It would be good to comment on this in the revised manuscript.

We do not have a proper explanation for the relative weakness of the signal associated to the first harmonic signal compared to the second harmonic. Unfortunately we do not have a proper set of experimental data to try to tackle this question.

At a speculative level, any nonlinearity induced by gate modulation can lead to transitions induced by two photons. Using the g-matrix formalism we developed in Ref. 23 to explain the EDSR mechanism, one can predict two-photon transitions by considering a second order

term for the gate dependence of the g-matrix: $\hat{g}(V) = \hat{g}(V_0) + \hat{g}'(V_0)\delta V + \frac{1}{2}\hat{g}''(V_0)(\delta V)^2$ with $\delta V = V - V_0 = V_{ac} \sin \Omega t$ being the microwave tone and Ω the Larmor frequency. The last term will induce spin transitions for gate-driven modulations at frequency $\Omega/2$.

Finally, as the first and second harmonic transitions occur at different magnetic fields, it is hard to assess experimentally the reasons behind the two signal strengths. The QD energy spectrum is different and the relaxation times as well; also, the exact microwave power delivered is different.

We have modified the text of the manuscript as follows (line 218), including the new reference 35 (Scarino et al, PRL 115, 106802 (2015)):

“Though both first and second harmonic excitations can be expected [35], the first harmonic EDSR line (inset to Fig. 3a) is unexpectedly weaker. A comparison of the two signal intensities requires the knowledge of many parameters (relaxation rate, microwave power, field amplitude) and calls for deeper investigations.”

6. In the discussion of the coherent spin control protocol on Page 4 and in Figures 3 & 4, I found it strange/confusing that they labelled the coherent control point as “M”, which to me indicates “Measure” point. It would be much clearer if they relabelled “M” as “C”, for “Control”. The point “I” can stay as “I”, or could be “I/M”, for “Initialize/Measure”.

We thank the Reviewer for the insight. Point “M” has become “C” (“Control”) and “I” now reads “I/R” (“Initialize/Read”).

7. Could the authors please comment on why the Rabi oscillations in Figure 4d have such a strange looking envelope? Why do they not appear to decay in a more typical manner?

We don't know the exact origin of the envelope of the Rabi oscillations. It might be related to a noise of the readout circuitry influencing the dispersive response. As we never observed such beating-like envelope in previous hole qubit Rabi oscillations (Refs. 6, 23 and unpublished data) from source-drain transport, we believe the noise responsible of the envelope is related to the reflectometry circuitry. For instance, given the dispersion in our chevron Rabi plot (fig. 4c), the mixing of f_C with f_R could lead to slow, incoherent rotations of the qubit, likely with an impact on the visibility of the Rabi oscillations. Also, the cryo amplifier can increase the noise at the readout stage via its backaction to the device gate.

In any case, we hope that technical improvements in the reflectometry circuitry and a better triggering of the rf tones could mitigate this effect in future experiments.

To comment the envelope of the Rabi oscillations, we have added the following sentence (line 299):

“The non-monotonous envelope is attributed to random phase accumulation in the qubit state by off-resonant driving at $f_{\text{Larmor}} \pm f_R$ due to up-conversion of microwave and reflectometry tones during the manipulation time.”

8. On a related note, in Figure 4d, why did they stop taking data at 1.2 us burst length? What does the data look like up to 2 microseconds? This would give a clearer idea of the decoherence

mechanism.

Data have been recorded only up to 1.4 μs burst length, as shown in the plot below. The visibility is further reduced from 1.2 μs to 1.4 μs , but without suggesting any evident functional form of the envelope.

9. Why did they not do a Ramsey sequence to measure T_2^ ? This seems an obvious experiment. Can they comment on this?*

Instead of characterizing the coherence time of the qubit, we preferred to advance the readout protocol by gating the acquisition window. Unfortunately, we had to warm up the cryostat before demonstrating any significant improvement in the visibility and have no time to perform any Ramsey experiment.

10. Do the authors believe that T_1 processes ultimately limit coherence in this experiment? If so, they should also state this.

We agree with the Reviewer that T_1 could be a candidate as limiting factor of coherence. This was suggested by a Rabi experiment in a previous cooldown on the same device. However, for the working point reported in the submitted manuscript, we varied the position of the microwave burst within the pulse to the manipulation point. We observed no evident variation in the EDSR signal up to 12 μs between the burst and the end of the pulse. This yields a lower bound of the relaxation time of the single hole spin. Further investigations are on the way to clarify this important point.

The text has been modified with what reported in the reply to question 7 and with the new sentence below (line 287):

“By shifting the position of a 100 ns microwave burst within a 12 μs pulse, no clear decay of the dispersive signal is observed, which suggests a spin lifetime at manipulation point longer than 10 μs .”

Finally, I list here a few minor suggestions, typos or grammatical errors that I picked up:

We have implemented the suggestions of the Reviewer and fixed the typos.

Reviewer #2 (Remarks to the Author):

In the following we answer the questions from Reviewer 2.

Conceptually, the followed approach is very similar to the work of Reference 22 where a coplanar waveguide resonator was used instead of the lump-element resonator to achieve essentially the same, i.e. dispersive spin readout by spin blockade in a double quantum dot. This conceptual link with the state-of-the-art is not well explained, giving a somewhat false impression about the novelty of the work.

We apologize for not having stressed enough the link with the work cited by the Reviewer. We have inserted a new sentence at line 58 of the main text to better underline the relevance of Ref. 22:

“In a similar fashion, the phase shift of a superconducting microwave resonator coupled to the source of an InAs nanowire has enabled spin qubit dispersive readout [22].”

Emphasizing on this new understanding, could the authors offer a discussion on the prospect of single-shot readout for their unique spin qubit platform, i.e. what is missing to get there in terms of quality factor, monolithic integration of the lumped-element resonator, noise level, etc.?

We extend the discussion paragraph accordingly to the remarks by the Reviewer as follows (lines 332):

“Technical improvements intended to enhance the phase sensitivity, like resonators with higher Q-factor and parametric amplification, could push the implemented readout protocol to distinguish spin states with a micro-second integration time, enabling single shot measurement as reported in a recent experiment with a gate-connected superconducting resonant circuit [40]. Lastly, the resonator integration in the back-end of the industrial chip could offer the possibility to engineer the resonant network at a wafer scale, guaranteeing controlled and reproducible qubit-resonator coupling.

The gate-based qubit sensing demonstrated here does not involve local reservoirs of charges or embedded charge detectors. This meets the requirements of forefront qubit architectures (e.g. Ref. 16), where the spin readout would be performed at will by any gate of the 2D quantum dot array by frequency multiplexing.”

The authors may consider the following other questions in improving further the manuscript

1. Would spin-flip tunneling terms in the model Hamiltonian change the overall interpretation? If yes, why can these processes be neglected given that they are another possible effect of the spin-orbit coupling?

Spin-flip tunneling terms would add additional contributions to the total phase signal expected from the model. Considering the singlet-triplet basis of Equation S2 of Supplementary Material, the detuning position of the anticrossings of the polarized triplets $T+(1,1)$ and $T-(1,1)$ with the singlet $S(0,2)$ have a strong dependence on the magnetic field.

We found no evident signal associated to such transitions in the detuning vs B field scans at the B-field orientations discussed in the main text. In other words, even if such matrix elements are not zero, they must be so small that the associated dispersive signal is below our detuning resolution.

The same can be said about the hybridization of $T_0(1,1)$ with $S(0,2)$. Still, this spin-flip term is more subtle as it leads to a dispersive response close to the zero detuning point. In the following, we refer to the levels in Fig. 2d and related colors, and name t_0 the tunnel coupling between $T_0(1,1)$ and $S(0,2)$. From simulations, we observe a two-fold effect in the DQD spectrum when t_0 is comparable with the bare (i.e. between $S(1,1)$ and $S(0,2)$) tunnel coupling t : i) at positive detuning, the repulsion between the blue and the black state is enhanced; ii) at negative detuning, the energy gap between the green and the blue state is reduced. As a result, the phase resonance associated to the green state (peaked at $\epsilon < 0$) becomes sharper and stronger, while the resonance related to the blue state (centered at positive ϵ) gets broader and lower. Magnetospectroscopy data in Figs. 2 and 3 show that the double-peak feature consists in a main resonance with a well-visible shoulder on the right edge, consistent with calculations for $t_0 \ll t$.

We conclude saying that we can't rule out that such spin-flip tunneling terms might be sizable for other orientations of the external magnetic field B not investigated in the present study.

We have added the following discussion to the Supplementary Material in Supplementary Note 3 ("DQD dispersive response" section):

"In the basis set of Eq. S2, the spin-orbit (SO) transition matrix elements are supposed weak compared to t and the Zeeman terms. Sizable spin-flip tunnelling terms like $t_{\text{SO}}^{\text{SO}} \langle \text{ket}\{T_{-}\} \text{ket}\{T_{-}(1,1)\} \text{bra}\{S(0,2)\}$ and $t_{\text{SO}}^{\text{SO}} \langle \text{ket}\{T_{+}\} \text{ket}\{T_{+}(1,1)\} \text{bra}\{S(0,2)\}$ would lead to a dispersive signal with a strong magnetic field dependence. We found no evidence of the corresponding dispersive signals in the magnetospectroscopy data discussed in the main text. \\\

A coupling factor $t_{\text{SO}}^{\text{SO}} \langle \text{ket}\{T_0(1,1)\} \text{ket}\{T_0(1,1)\} \text{bra}\{S(0,2)\}$ between $T_0(1,1)$ and $S(0,2)$ comparable to t has not to be expected neither. From simulations at $B > 0.5 \text{ T}$, with $t_{\text{SO}}^{\text{SO}} \langle \text{ket}\{T_0\} \text{ket}\{T_0\} \text{bra}\{S(0,2)\} \sim t$ the phase resonance of the interdot transitions would resemble a pronounced peak with a barely-visible shoulder on the right edge, not consistent with data in Figs. 2 and 3 of the main text.

As a final note, we can't rule out these such spin-flip tunneling terms might be relevant for orientations of the external magnetic field different from those investigated here."

2. The electronic temperature is stated to be 250 mK, corresponding to an energy of 22 micro electron-volt. How was the smaller tunneling coupling energy deconvoluted from the thermal broadening when using the data of Figure 2a? In usual charge sensing, the inter-dot charge transition always has a temperature broadening associated to it.

According to simulations at 0 Tesla of a $(1,1) - (0,2)$ interdot charge transition, the impact of T_{eff} on the dispersive signal by gate reflectometry is twofold. The amplitude of the phase shift diminishes by increasing T_{eff} . However, contrarily to an interdot signal probed by a charge detector, the full width at half maximum (FWHM) doesn't increase monotonously with T_{eff} . This is linked to the sensitivity of the resonator, ultimately constrained to the detuning region where the DQD states have finite curvature (if the phase signal is supposed composed uniquely of quantum capacitance contributions).

In the revised manuscript we have added Supplementary Information (Note 3) to clarify this concept. Supplementary Figure 3 shows the evolution of the FWHM of the interdot phase

signal as a function of the thermal energy $k_B T_{\text{eff}}$ normalized by the tunnel coupling t . The FWHM values are in units of t .

Here the additional information reported in Supplementary Information (Note 3):

“Two limiting situations are envisaged. At low temperature, $k_B T_{\text{eff}} < t/10$ and the width of the interdot signal is set by the tunnel coupling to $\sim 3t$. Here just the ground singlet is populated. In the opposite limit of high temperature, $k_B T_{\text{eff}} > 2t$, the threefold triplet and both bonding and anti-bonding singlet are thermally populated; by sweeping T_{eff} , the magnitude of the interdot resonance drops, but the FWHM saturates at $\sim 4t$. In the intermediate regime, the FWHM increases progressively with T_{eff} , up to the saturation point occurring at $k_B T_{\text{eff}} \simeq 2t$.

Furthermore, Supplementary Figure\,\ref{fig:Fig3S} demonstrates that the FWHM allows to estimate t in the $(3t, 4t)$ range whatever the temperature is. This distinguishes dispersive readout from charge sensing (especially when $k_B T_{\text{eff}} > 2t$), as the resonator sensitivity is ultimately constrained to the avoided crossings in the energy level diagram.

Fits to the interdot detuning phase shift yield $t=8.5\,\mu\text{eV}$ and $t=6.4\,\mu\text{eV}$ in the low and high temperature limit, respectively. The evolution of the interdot transition line versus the magnetic field is reproduced qualitatively assuming the lowest tunnel coupling and $0.25\,\text{K}$ as effective temperature.”

Coming to the point arisen by the Reviewer, the stated interdot tunnel coupling (7 ueV) has been extracted from the FWHM of the interdot transition line at 0 Tesla from $t = (t^{\text{HIGH}} + t^{\text{LOW}})/2$, where t^{HIGH} (6.4 ueV) and t^{LOW} (8.5 ueV) have been evaluated in the high T and low T limit, respectively.

We realize now that this claim is maybe too simplistic. We have amended the paragraph in the main text as follows (line 127):

“We estimate t between 6.4 and 8.5 μeV , depending on whether thermal fluctuations contribute or not to the dispersive response (see Supplementary Note 3).”

3. I can appreciate the hard work involved but why not showing the EDSR resonance of the other dot?

We didn't manage to find a condition for which both the EDSR resonances were visible and of publication quality. Yet we report here a map $|B|$ vs microwave frequency for the orientation of the field used in the qubit dataset shown in the manuscript. A 90 ns microwave burst was applied during a pulse of 250 ns with a 1/3 duty cycle. Two lines can be distinguished, marked as A and B. Line A shows the strongest dispersive signal and corresponds to spin flips in the left dot; line B is related to spin flips in the right dot.

4. In what conditions removing the reservoirs would enable higher operating temperature? This statement is repeated twice but not explained nor referenced.

Following the remark of the Reviewer, we discuss this point and we refer explicitly to the work of Ref. 19. A new sentence appears in the last paragraph of the paper (lines 350):

“Dispersive spin detection by Pauli blockade has a fidelity not constrained by the temperature of the leads. As recently shown [42], isolated DQDs can serve as spin qubits even if placed at environmental temperatures exceeding the spin splitting, like 1K or more. This should relax many cryogenic constraints and support the co-integration with classical electronics, as required by a scale-up perspective [19].”

Reviewer #3 (Remarks to the Author):

Here the answers the questions from Reviewer 3.

I have some detailed technical comments:

- Both the abstract and the conclusion claim that operation “may be extended well above 1 Kelvin”. This claim should be justified or removed, especially since the authors apparently decided not to test it directly by warming their cryostat. Anyone copying this technique would

no longer need to worry about the reservoir temperature, but the readout might still be affected by excited charge states, phonons, other spin relaxation mechanisms, etc.

We agree with the Reviewer that dispersive spin readout by spin blockade removes one of the bottlenecks for qubit operability at high temperature, but other challenges have to be tackled. Encouraging results are provided by a very recent work by C. Yang et al., arXiv 1902.09126 (2019) reporting gate fidelities and coherence times comparable to qubits operated at millikelvin temperatures. This paper is now referenced as Ref. 42 and new comments now appear in the conclusions (line 350):

“Dispersive spin detection by Pauli blockade has a fidelity not constrained by the temperature of the leads. As recently shown [42], isolated DQDs can serve as spin qubits even if placed at environmental temperatures exceeding the spin splitting, like 1K or more. This should relax many cryogenic constraints and support the co-integration with classical electronics, as required by a scale-up perspective [19].”

- The magnetic field direction in Fig. 2 is not clear.

We have added the magnetic field direction in the caption of Fig. 2b.

- The second harmonic driving process in Fig. 3 is indeed surprising. Is it possible that the strong line is in fact the fundamental? I do not know whether such a large g-factor is plausible in this material. There is a paper (PRL 115, 106802) that shows how to tell by measuring the phase dependence in a Ramsey experiment.

We thank the Reviewer for the reference suggested. We did not perform a Ramsey experiment to clarify the nature of the measured harmonics. Still, the two EDSR lines of Fig.3a (main and inset) fulfill the conditions $2hf = g\mu B$ and $hf = g\mu B$, respectively, sharing the same g-factor.

Moreover, from previous experiments on hole quantum dots we expect a g-factor in the range 2 ± 0.7 (Voisin et al, Nano Lett. 16, 88 (2016) and Refs. 6, 23). Possible explanations about the driving mechanism of the second harmonic has been given to the question 5 from Reviewer #1.

-Some small errors...:

We have amended the typo and corrected the text and figure, as suggested.

Note: We have also amended a typo in the caption of Figure 1. The width of the Si channel is not 15 nm but 35 nm.

Alessandro Crippa,

on behalf of all co-authors.

REVIEWERS' COMMENTS:

Reviewer #1 (Remarks to the Author):

It appears to me that in this revised version of the manuscript the authors have taken on board all of the comments that I made in my review, together with those of the other two referees. I think that this addresses all of the outstanding issues in the original manuscript, and that the revised manuscript is now suitable for publication in Nature Communications.

Reviewer #2 (Remarks to the Author):

The authors addressed all comments and suggestions from my initial assessment. I strongly recommend this work for publication in Nature Communications.

Reviewer #3 (Remarks to the Author):

The authors have satisfied my concerns and I recommend publication.